# Peatland Fire Weather Conditions in Central Kalimantan, Indonesia

Aswin Usup [1,2,*] and Hiroshi Hayasaka [3]

1   Faculty of Agriculture, University of Palangka Raya, Palangka Raya City 74874, Indonesia
2   Research Center for Fire Prevention and Land Rehabilitation, Research Institute and Community Service, University of Palangka Raya, Palangka Raya City 74874, Indonesia
3   Arctic Research Center, Hokkaido University, Sapporo 0010021, Japan
*   Correspondence: aswin@agr.upr.ac.id; Tel.: +62-8115249911

**Abstract:** Peatland fires in Central Kalimantan emit thick smoke and large amounts of greenhouse gases and have an impact on the environment globally, but studies on fire weather have not been carried out due to lack of diurnal weather data. The aim of this study is to identify the fire weather conditions during active fires that is needed to mitigate future occurrences of peat fires in Indonesia. The available diurnal weather data was used to analyze the fire weather conditions. Based on meteorological data on active fires (11 days), there was a significant increase in air temperature due to the sea breeze that started blowing in the morning. The average values for the 11-day period around 15:00 are a maximum air temperature of 36 °C, minimum humidity of 37%, wind speed of 21 km h$^{-1}$, and a rate of increase of 2.7 °C h$^{-1}$ from 8:00. The difference in sea and land temperatures causes strong winds to blow and triggers an increase in land temperatures. The results of this report can help predict fire activity at high temperatures in the future based on global warming predictions made by other researchers. The rapid rate of increase in air temperature from the morning will be useful in anticipating fires in Central Indonesia.

**Keywords:** peatland fires; diurnal weather conditions; outgoing longwave radiation (OLR); sea breeze; El Niño-southern oscillation (ENSO); Mega Rice Project (MRP); sea surface temperature (SST); ground water level (GWL)



## 1. Introduction

Indonesia has one of the highest rates of deforestation and forest degradation in the world. The main causes are agricultural expansion and timber extraction as well as increasing incidence of forest fires [1–4]. Indonesian peatlands store an estimated 57 Gt of carbon, 55% of the world's tropical peatland carbon [5,6]. Indonesia's peatlands have been extensively degraded and peatland fires are on the rise. Especially in Central Kalimantan (Borneo), where large-scale agricultural land development has taken place, peatland fires occur annually. The largest development one is Mega Rice Project (MRP) [7]. Recurring peatland fires throughout Central Kalimantan have resulted in severe economic and social impacts for local people along with globally significant environmental impacts.

Under climate change, Indonesia is predicted to experience temperature increases of approximately 0.8 °C by 2030 and will exceed 1.5 °C in the near term (2021–2040) [8]. Moreover, rainfall patterns are predicted to change, with the rainy season ending earlier and the length of the rainy season becoming shorter [9]. Warming temperatures and rainfall trends in Indonesia have been reported to support this prediction. Estimates from the Berkeley Earth dataset suggest that annual mean temperatures in Indonesia were typically about 0.8 °C above the 1951–1980 baseline for the 2010–2017 period. Since 1960, hot days and nights have increased by 88 days and 95 nights per annum, respectively, especially during the summer months of July–September [10]. Studies have noted an overall decrease in average annual precipitation [11]. Rising sea temperatures could increase the frequency

of El Niño events and intensify droughts [12]. Thus, fires in Indonesia will be more active under warmer conditions in near future.

Nevertheless, fire weather in Indonesia has not been clarified during an active fire. There are few papers on fire weather. Most papers have explained activities of fire using rainfall data to evaluate drought conditions related to groundwater level (GWL) and the El Niño-Southern Oscillation (ENSO). A. Sulaiman et al. analyzed the teleconnected relationship between groundwater levels (GWL) and extreme climatic conditions, such as ENSO and positive Indian Ocean Dipole (IOD+). They showed that the dropped sea surface temperature anomaly induced by anomalously easterly winds along the southern Java-Sumatra occurs several weeks before the GWL drop to the lowest value [13]. Another study using rainfall data in Kalimantan proposed that seasonal precipitation forecasts should be central to Indonesia's evolving fire management policy derived from their analysis of long, up-to-date series observations on the burnt area, rainfall, and tree cover [9]. These studies only indirectly assess fires using mainly rainfall data.

In this report, we discuss fire weather conditions during an active fire period based on analysis results using fundamental weather parameters of air temperature, humidity, air pressure, wind speed, and direction. However, as Indonesia does not have long-term reliable meteorological data, we have to use the data on the Internet. Only one site in central Kalimantan has diurnal weather data (air temperature, relative humidity, wind speed, and wind direction) of every 30 min from 12 am to 11:30 pm. This report's results may help to assess future fire activities under higher temperatures.

## 2. Materials and Methods

### 2.1. Study Area

The study area in Kalimantan (Borneo) covers the latitude range from 1.5° to 3.5° S (south latitude) and the longitude range from 113° to 115° E (area is about 49,500 km$^2$) as shown in Figure 1. Palangka Raya (2.21° S, 113.92° E) is the capital of Central Kalimantan and is located in the southern part of the province. Banjarbaru airport weather station (3.45° S, 114.74° E) near Banjarmasin is located in the southeastern part of the study area.

Kalimantan has approximately 57,600 km$^2$ of peatlands with 30,100 km$^2$ in Central Kalimantan alone. Figure 1 shows the main part of the peatland in Central Kalimantan. The distance from the nearest coastline is about 100 km and the average altitude is only around 10 m above sea level. The MRP was built on tropical swamp forest areas on the eastern and southern sides of Palangka Raya. Before the disturbance, the tropical swamp forest could hold enough water to stay wet even in the dry season. However, the 4000 km long MRP canal built for irrigation facilitated not only illegal logging but also a loss of water through drainage from most of the peatlands in the MRP area. These disturbances are the main reasons for intense and recurrent fire activity in the MRP area.

We refer to the study area as 'MRP+e' in this report as our previous report already defines 'MRP+' (1.75–3.5 S, 113.5–115.0 E, area is about 32,400 km$^2$, smaller cover area than 'MRP+e') [14]. MRP+e covers the MRP area and its vicinity (a part of the Sebangau National Park) and was chosen as the study area simply because MRP+e area has major peatland in Central Kalimantan and is one of the highest hotspot density areas in Indonesia [14].

The previous paper about peatland fires in 'MRP+' area [14] reported: The area of 'MRP+' (about 32,400 km$^2$) accounts for about 6% of Kalimantan Island (544,150 km$^2$). Nevertheless, about 20% of all Kalimantan fires have occurred in the study area. This suggests that peatland fires existing in MRP+ contribute greatly as the total number of HSs of top seven fires in MRP+ was 77,446, about 80% of the total number of HSs (97,423) for 20 years from 2002.

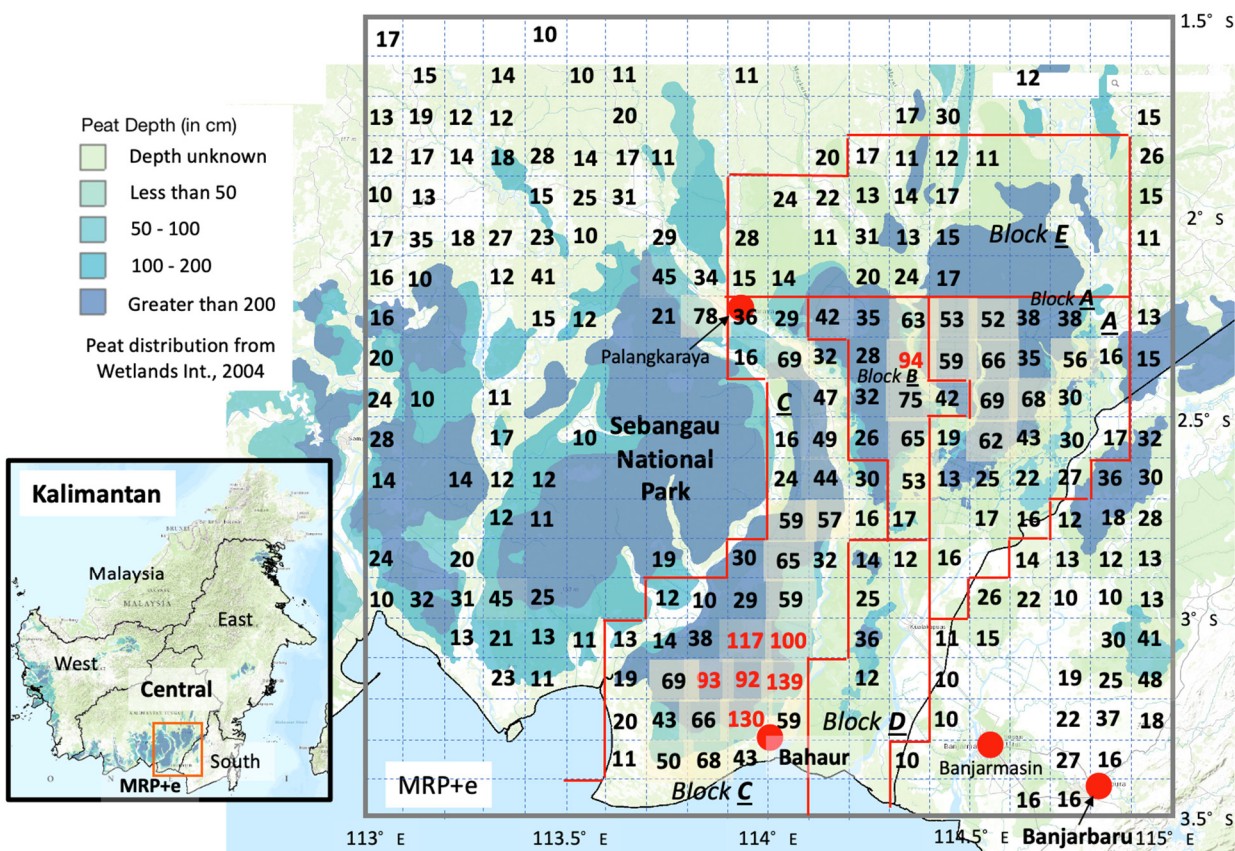

**Figure 1.** Central Kalimantan (Borneo) in Indonesia. Fire distribution is shown by number of hotspots (HSs) for each grid cells (latitude and longitude, 0.1° × 0.1° resolution). Mega rice project (MRP) cover area: 1.5–3.5 S, 113–115 E, and boundary of five block A to E are shown roughly by red line. The top seven fire-prone areas are indicated by red numbers. Based map made by "esri".

### 2.2. Fire Hotspot (Hotspot) and Weather Data

Hotspot (HS) data of 21 years from 1 July 2002 to 31 December 2022 detected by Moderate Imaging Resolution Spectroradiometer (MODIS) on the Terra and Aqua satellites were used to evaluate wildfires in Kalimantan. MODIS HS data were obtained from the NASA Fire Information for Resource Management System https://firms.modaps.eosdis.nasa.gov/download (accessed on 15 January 2023) with a pixel resolution of 1 km. The number of daily HSs is used to identify fire season, active fire period, and the dates of major HS peaks during the fire periods.

Various daily weather maps, such as pressure, wind, temperature at 500 hPa, 850 hPa and 925 hPa, sea surface temperature (SST), obtain each of those anomalies from the NCEP/NCAR 40-year reanalysis data (https://psl.noaa.gov/data/composites/day/, accessed on 15 January 2023). We analyzed these to find fire-weather conditions during active fire-\ periods, fire-related synoptic-scale circulation patterns, rainy conditions, STT, etc. Coverage and spatial resolution of the NCEP reanalysis data are geographic longitude and latitude: 0.0°–358.125° E, 88.542°–88.542° N; spatial resolution: about 2.5° × 2.5°; period and temporal resolution: 1 January 1948 to present. Long-term means (climatologies) are based on 1991–2020.

Daily rainfall data for Bahaur area (set at 3.2° S, 114.1° E) for 21 years from 2002 to 2022 are obtained from the JAXA (https://sharaku.eorc.jaxa.jp/GSMaP_CLM/index.htm, accessed on 15 January 2023). The JAXA rainfall data are generated with information from multiple precipitation-observing satellites and geostationary meteorological satellites.

Diurnal weather data (temperature, relative humidity, wind speed, and wind direction) of every 30 min from midnight (12 AM) in Banjarbaru airport weather station at 3.45° S,

114.74° E are obtained from the Weather Underground (https://www.wunderground.com/history/daily/id/banjarbaru/WAOO/date/2015-9-15, accessed on 15 January 2023).

Niño 3.4 data (region: 5° N–5° S, 170°–120° W, one of the ENSO indices) is obtained from the NOAA web site (https://www.cpc.ncep.noaa.gov/data/indices/, accessed on 15 January 2023).

Outgoing longwave radiation (OLR an NOAA) measured by the Atmospheric Infrared Sounder (AIRS) on EOS Aqua (the second Earth Observing System polar-orbiting platform) is obtained through "EOSDIS Worldview (NASA)" (https://worldview.earthdata.nasa.gov, accessed on 26 February 2023) used to show their correlation with fires in Central Kalimantan.

Outgoing longwave radiation (OLR–MC, JMA) is a rain-related index derived from outgoing longwave radiation (OLR) issued by Japan Meteorological agency (JMA). (http://www.data.jma.go.jp/gmd/cpd/db/diag/emi/emi.html, accessed on 26 February 2023).

### 2.3. Analysis Methods

MODIS hotspot (HS) data of 21 years (2012–2021) were used to clarify the spatiotemporal structure of peatland fires in Central Kalimantan. The distribution of fires was determined by setting the 0.1 degrees grid cell (0.1° × 0.1°, latitude and longitude) as shown in Figure 1. Daily hotspot (HS) data were used to show fire trends, fire activities, annual fire history, fire-prone areas, fire season, and active fire-period in Central Kalimantan.

Daily rainfall data of 21 years (2012–2021) at Bahaur were used to define dry and wet seasons, and rainfall trend. Rainfall pattern in 2015 and 2021 were used to explain drought conditions in 2015 and wet conditions in 2021.

Fire activities were evaluated by considering ground water level (GWL). Three types of peatland fire are surface fire (GWL = 0 to −300 mm), shallow peatland fire (GWL = −300 to −500 mm), and deep peatland fire (GWL = lower than −500 mm) [14,15].

Weather data (temperature, relative humidity, wind speed, and wind direction) from every 30 min from midnight (12 AM) in Banjarbaru were used to diurnal weather conditions during the active fire-period in 2015 and La Niña weather conditions in 2021.

Analysis using various synoptic scale weather maps at various air levels (1000, 925, 850, and 500 hPa) were also used to grasp fire-weather conditions during the active fire periods.

The NOAA composite weather maps were used to identify the presence and location of high pressure, low pressure, sea surface temperature, wind speed and direction, wind vector, and their anomalies (1991–2020 climatology).

Niño 3.4, Sea Surface Temperature Anomaly (SSTA), Outgoing longwave radiation (OLR-NOAA), and Outgoing longwave radiation (OLR–MC) were used to show their correlation with fire activities in Central Kalimantan.

## 3. Results

### 3.1. Fire Distribution

Fire spatial distribution in the study area is shown in Figure 1. The number of each grid cell (0.1° × 0.1° resolution) is the average number of HSs in last 21 years. Grid cells without numbers show weak fire areas where the average number of HSs are less than 10. A large average number of HSs (>100) or frequent fire areas shown by light yellow rectangles are found in the southern part of the study area (the southern MRP Block-C). On the contrary, weak fire areas (HSs < 10) are found mainly in Sebangau National Park, MRP Block E, and forest areas.

The highest number of HSs is 139 in northern Bahaur. As with the other three high HSs grid cells also located in the southern MRP Block-C, we analyzed rainfall and weather conditions in the southern areas of central Kalimantan in the following sections.

### 3.2. Fire Months and Rainfall

Figure 2a shows the fire history in the study area (MRP+e) from 2002 to 2022. The annual number of hotspots (HS) for each year is the cumulative daily HS for each year. The

total HSs of the 21 years is 139,939 and the average annual HSs is 6664 (the rightmost bar graph in Figure 2a). The largest number of HSs (about 24,000) in 2015 occurred under El Niño conditions (Niño 3.4, annual mean SSTA is +1.4). One of the smallest HSs (=91) in 2021 is related to La Niña (Niño 3.4, annual mean SSTA is −0.6 in Niño 3.4). The years with more than twice the average HSs are called "Fire Year" in this report. The top five fire years are 2002, 2006, 2008, 2015, and 2019.

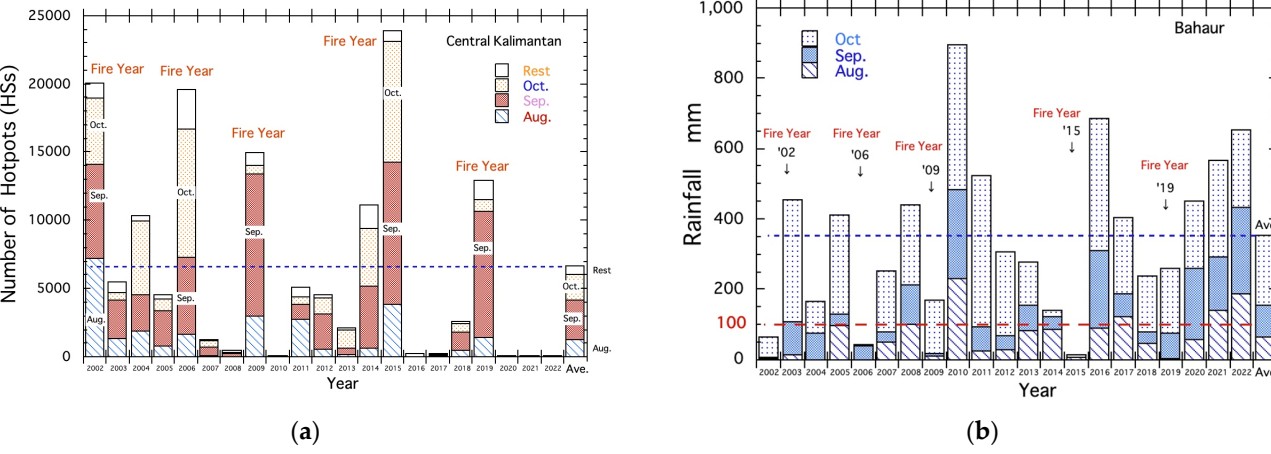

(**a**)                                     (**b**)

**Figure 2.** (**a**) Fire history in Central Kalimantan. (**b**) Rainfall of fire three months (August, September, and October).

Each annual bar graph in Figure 2a is divided into four to show monthly HSs. Those from the bottom to the top are HSs in August, September, and October, and the rest of the HSs. Figure 2a clearly shows most fires occurred in the three months of August, September, and October. The average bar graph in Figure 2a shows fire and the top month is September, October follows, and August is third.

Large differences in HSs among "Fire Years" and small fire years, such as 2010, 2020, 2021, and 2022 are related to rainfall in August, September, and October. Figure 2b shows rainfall in weak fire years exceeding the average rainfall (350 mm). On the contrary, rainfall amounts of five "fire Years" are less than 100 mm. Drought months for the top five fire years are shown by A, S, and O, the capital letter of three months. Especially, rainfall of the top fire year, 2015, is only 13.2 mm. Drought conditions in 2015 were the most severe among last 21 years.

### 3.3. Average Dry and Fire Season

Figure 3a shows the average dry and rainy season in southern Central Kalimantan. Two seasons are defined by gradients of red thin straight lines on the accumulated rainfall curve (thick green curve). The dry season lasts 164 days from 10 May to 20 October with an average rainfall of about 4 mm during this period. The rainy season is from November to April with an average rainfall of about 11 mm. A very low rainfall period (average rainfall of 1.4 mm) in the middle of the dry season in Figure 3a may be one of the important weather conditions for very active fires in southern Central Kalimantan.

Figure 3b shows the average daily number of hotspots (HSs) and the accumulated HSs curve. Fire season from June is defined by five gradients of green thin straight lines on the accumulated rainfall curve (thick blue curve). The fire season starts on 16 June and lasts until 22 November (about 5 months, 160 days). The fire season, which begins about a month later than the dry season, suggests the need for vegetation and surface drying. We already clarified that fire periods are related to groundwater level (GWL) [14,15]. They are so-called surface fires ((SF), GWL = 0 to −300 mm), shallow peatland fires ((SPF), GWL = −300 to −500 mm), and deep peatland fires ((DPF), GWL = lower than −500 mm).

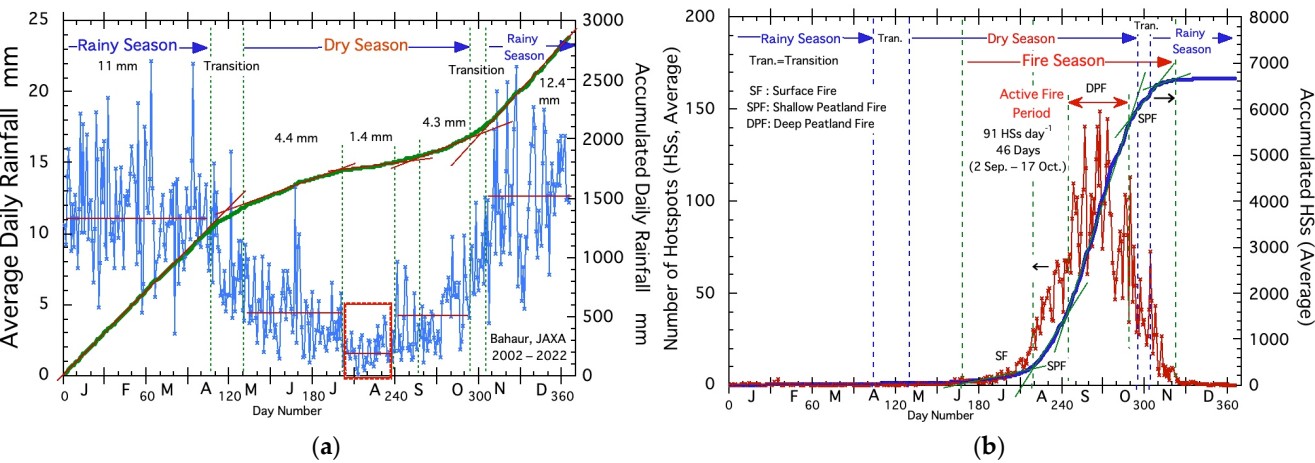

**Figure 3.** (**a**) Average dry season in southern Central Kalimantan. (**b**) Average fire season in Central Kalimantan.

In the first fire period, GWL decreases with the onset of the dry season and surface fires occur. Secondly, shallow peatland fires follow. Finally, deep peatland fires occur from the top of September as shown in Figure 3b. Deep peatland fires last from 2 September through 17 October (46 days, 91 HSs day$^{-1}$) and accounts for about 63% of annual average HSs. Thus, the third fire period is called the "active fire period" in this report and fires are very active due to deep peatland fires. After the "active fire period", as the GWL rises with the onset of the rainy season, shallow peatland fires and surface fires occur in the fourth and fifth fire period, respectively.

### 3.4. Various Fire Weather Conditions

From here, we discuss various fire weather conditions in 2015 under El Niño (top fire year) and 2021 under La Niña (19th (weak) fire year).

### 3.4.1. Rainfall and Fire in 2015 and 2021

Figure 4a shows daily rainfall trend in 2015 (El Niño year) and 2021 (La Niña year). In 2015, the drought started from 26 June and lasted until 28 October (125 days, about 4 months). Total rainfall during this period was only 19.4 mm (0.15 mm day$^{-1}$). In comparison, considerable rainfall (5.9 mm day$^{-1}$) during in the middle of the dry season occurred in 2021 as shown in Figure 4a.

Figure 4b shows the fire occurrence trend in 2015. The total number of HSs was 23,867 and about 3.6 times the average (6664). The active fire period in 2015 started on 30 August and lasted until 26 October. The fire rate during about two months (58 days, September and October) was 343 HSs day$^{-1}$. This rate is about 3.8 times larger than the average HSs rate (91 HSs day$^{-1}$ in Figure 3b). As fires in 2021 were very weak and the total number of HSs was 91, it is indicated by a blue round circle in the lower right corner of Figure 4b.

Various weather conditions (air temp., relative humidity, wind speed, and directions) from 12 to 22 September (one of during active fire periods in 2015) are summarized in the next section.

### 3.4.2. Weather Conditions in 2015 and 2021

To find fire-prone weather conditions, we used weather data every 30 min from midnight (12 AM) from the 12 to the 22 of September measured at Banjarbaru. This period contains the top and second largest HSs peak day (1285 and 1029 HSs on September 15 and 22, respectively) in 2015. The total number of data was 528 (=11 days × 48 (every 30 min)). Data for the same period in 2021 are used for comparison. Unfortunately, there was a considerable number of missing data. For example, a number of missing data on temperature was 109 (about 20%). Especially, most of the data on 15 September (HSs peak

day) were missing (about 77%). In this section, diurnal changes in average temperature, relative humidity, wind speed, and wind direction during 11 days from 12 September are analyzed to clear fire weather conditions.

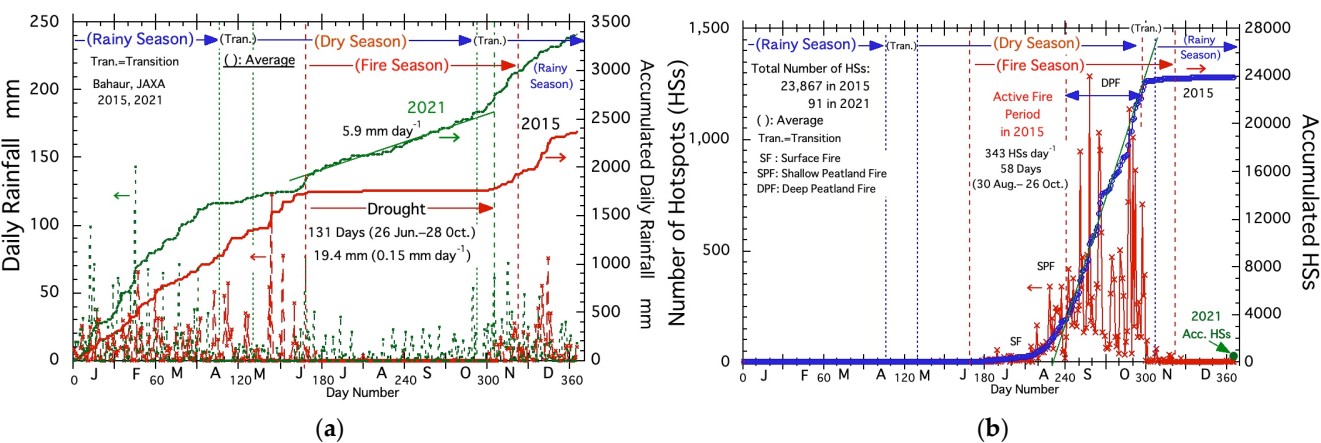

**Figure 4.** (**a**) Rainfall trend in 2015 (red) and 2021 (green). (**b**) Fire activities in 2015 (red and blue) and 2021 (green).

1. Diurnal change of temperature and humidity

Figure 5a,b show average air temperature and average relative humidity (RH) in 2015 and 2021, respectively. The x-axis is the time of day. Figure 5 shows the average maximum air temperature in 2015 was 36 °C at 14:30 and higher than that of about 31 °C at 14:30 in 2021. There is a relatively large difference in average RH in 2015 and 2021. A total of 37% at 14:00 in 2015 in Figure 5a is considerably lower than that of about 60% in 2021 in Figure 5b.

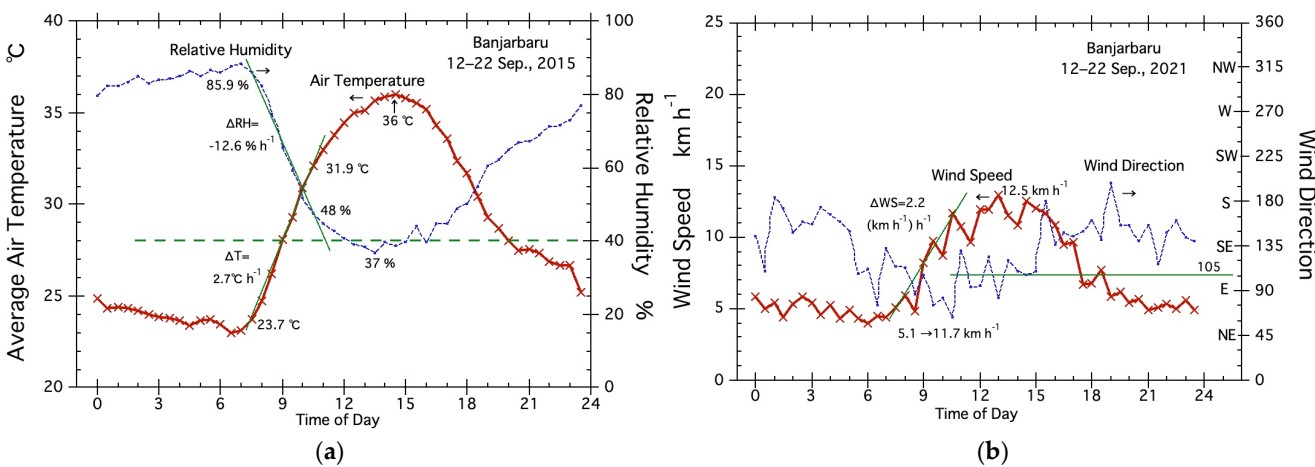

**Figure 5.** (**a**) Air temperature and relative humidity during 12–22 September 2015. (**b**) Same items in 2021.

In addition to these differences, there are also differences in the temperature increase rate (ΔT) and RH decrease rate (ΔRH) from 7:30 to 10:30 in the morning as indicated by the auxiliary lines in Figure 5. ΔT and ΔRH in 2015 are 2.7 °C h$^{-1}$ and −12.6% h$^{-1}$. Both values are larger than the corresponding values of 1.6 °C h$^{-1}$ and −8% h$^{-1}$ in 2021. After these large morning changes, air temperature rose another 4.1 °C to 36 °C and RH dropped below 40% in 2015. In contrast, air temperature in 2021 did not rise much and remained at 31.1 degrees, and RH was over 60%.

2. Diurnal change of wind speed and direction

Figure 6a,b show average wind speed and direction. The average wind speed and direction in Figure 6a shows maximum wind speed of southeasterly wind in 2015 is 21 km h$^{-1}$ (classification speed of moderate breeze, >19 km h$^{-1}$) at 17:00. On the contrary, the wind speed during the daytime (10 am to 4 pm) in 2021 was not so strong, mostly about 10–13 km h$^{-1}$ and mostly easterly wind. From those relatively large differences in wind speed, we may say strong wind conditions are one of the important fire weather conditions in central Kalimantan.

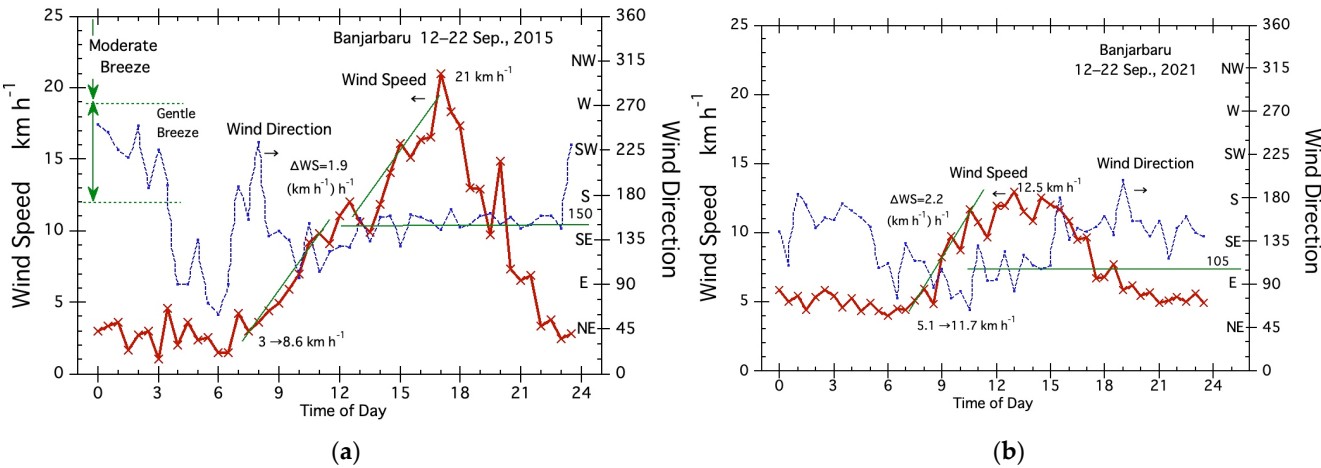

**Figure 6.** (**a**) Wind speed and direction during 12–22 September 2015. (**b**) Same items in 2021.

In addition to these differences, there are also differences in wind speed increase rate (ΔWS) from 7:30 to 10:30 in the morning as indicated by the auxiliary lines in Figure 6. ΔWS in 2015 and 2021 are 1.9 and 2.2 (km h$^{-1}$) h$^{-1}$. After the morning wind speed increases, wind speeds in 2015 continued to increase in strength, reaching a maximum speed of 21 km h$^{-1}$ at 17:00. On the other hand, wind speeds in 2021 did not increase much and were weakened around 15:00. The increase in wind speed in 2015 can be said to be a feature of the sea breeze which intensifies mainly due to the large temperature difference between land and sea. Since the sea surface temperature is almost the same in both years (301° K and 302° K on 22 September), it can be said that the land temperature is the main factor.

### 3.4.3. Diurnal Weather Conditions on Second HSs Peak Day

The weather data for 15 September 2015 (the top hotspot peak day) has a lot of missing data, so we analyzed fire weather conditions for the second HS peak day. Various diurnal weather conditions on 22 September 2015 are shown in Figure 7. From Figure 7, fires on this day became very active under a maximum temperature of 37 °C, minimum low humidity of 30% (Figure 7a), and maximum wind speed 28 km h$^{-1}$ (Figure 7b). Especially, moderate breeze (classification speed, >19 km h$^{-1}$) continued to blow from 13:00 to 19:00. Furthermore, air temperatures above 35 °C continued from about 10:00 to 17:00 pm. Relative humidity under 40% lasted from around 10:00 to 18:00. The wind direction was mostly easterly. From these diurnal weather conditions, we may say fire weather conditions in Kalimantan are Moderate breeze (>19 km h$^{-1}$), High air temperature (>35 °C), Low relative humidity (<40%), and Easterly wind.

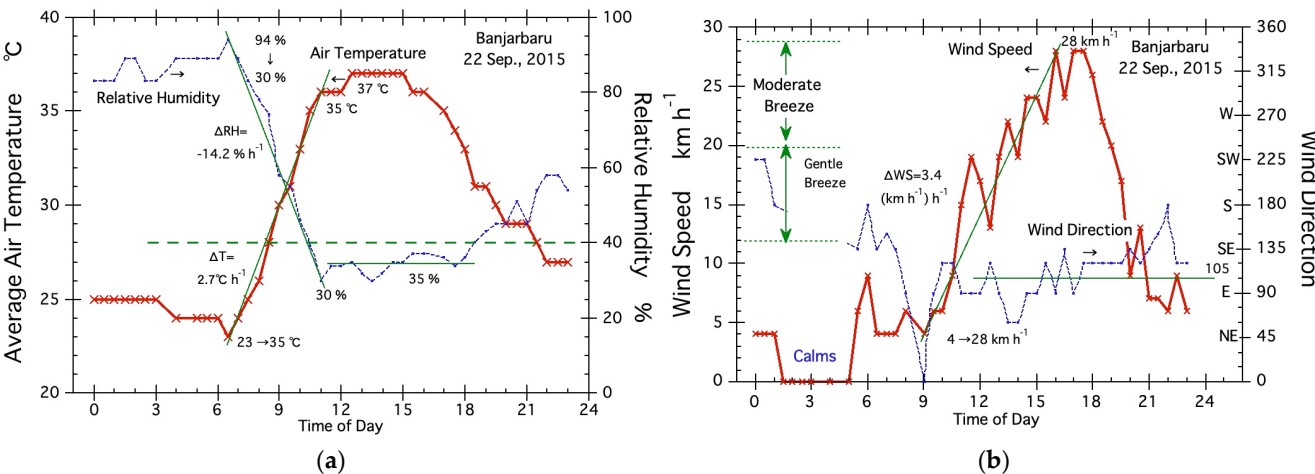

**Figure 7.** (**a**) Air temperature and relative humidity on 22 September 2015. (**b**) Wind speed and direction on 22 September 2015.

Temperature increase rate ($\Delta$T) and humidity decrease rate ($\Delta$RH) from 6:30 to 11:00 in Figure 7a were 2.7 °C h$^{-1}$ and −14.2% h$^{-1}$. Those values are not so different from their average values in Figure 5. The large difference is found in wind speed increase rate ($\Delta$WS). Wind speed increase rate ($\Delta$WS) from 9:00 to 16:00 on September 22 was 3.4. This rate is about two times higher than the average rate of 1.9 (km h$^{-1}$) h$^{-1}$ in Figure 6a. This suggests active fires on hotspot peak days occurred mainly due to the strong wind speed of the easterly wind.

### 3.5. Various Fire Related Index

Until 2018, the El Niño–Southern Oscillation (ENSO) was used as an explanation for fires in Indonesia's peatlands. However, when the 2019 fires occurred independently of El Niño, more suitable indicators and methods are required to (a) analyze, (b) evaluate, and (c) forecast peatland fires. The correlations of the three indices with fire activity were examined. Three indices are Niño 3.4 (region: 5° N–5° S, 170°–120° W, one of the ENSO indices), outgoing longwave radiation (OLR, NOAA), and outgoing longwave radiation (OLR–MC, JMA).

Niño 3.4 is one of the ENSO indices and is commonly used to describe fires in Indonesia. Figure 8a shows the relationship between the number of hotspots (HSs) and Niño 3.4 in September (active fire month). Niño 3.4 has a good correlation with HSs in most active fire years, such as 2015, 2002, 2006, and 2014. In comparison, Niño 3.4 could not explain active fires in 2019 (neutral year). From this large difference from the regression curve in 2019 and other years, the value of the decision coefficient ($R^2$) was 0.578 (the regression equation: HSs = 2455.1 + 3468.7 (Niño 3.4) + 826.39 (Niño 3.4)$^2$, $R^2$ = 0.57771).

To explain active fires in 2019, we first introduced outgoing longwave radiation (OLR–MC, JMA) in our study in 2020 [14]. OLR–MC, JMA is originally developed by JMA to monitor convective activity in Indonesia or Maritime Continent (MC). This rain-related index is derived from the OLR. In this study, the value of the decision coefficient ($R^2$) from the regression curve for OLR–MC, JMA was 0.823 (the regression equation: HSs = 2878.4 − 2993.8 (OLR–MC) + 500.6 (OLR–MC)$^2$, $R^2$ = 0.82298) and considerably higher than the above 0.578 (Niño 3.4).

In addition to OLR–MC, JMA, we introduce outgoing longwave radiation (OLR) from NOAA to evaluate land temperature in any place in Kalimantan. OLR is one of the indicators to evaluate rainfall cloud activity and is used to evaluate rainfall [16,17]. In this report, OLR is introduced as an indicator for evaluating ground surface temperature. We extract OLR at near Banjarmasin in southern central Kalimantan through "EOSDIS Worldview (NASA)". Figure 8b shows the correlation between OLR, NOAA and HSs. The value of the decision coefficient (R2) was 0.827 (the regression equation: HSs = 3.2111 × 10$^6$

$-$ 22,393 (OLR) + 39.038 (OLR)$^2$, R$^2$ = 0.82684) and was a little bit higher than that of OLR–MC, JMA.

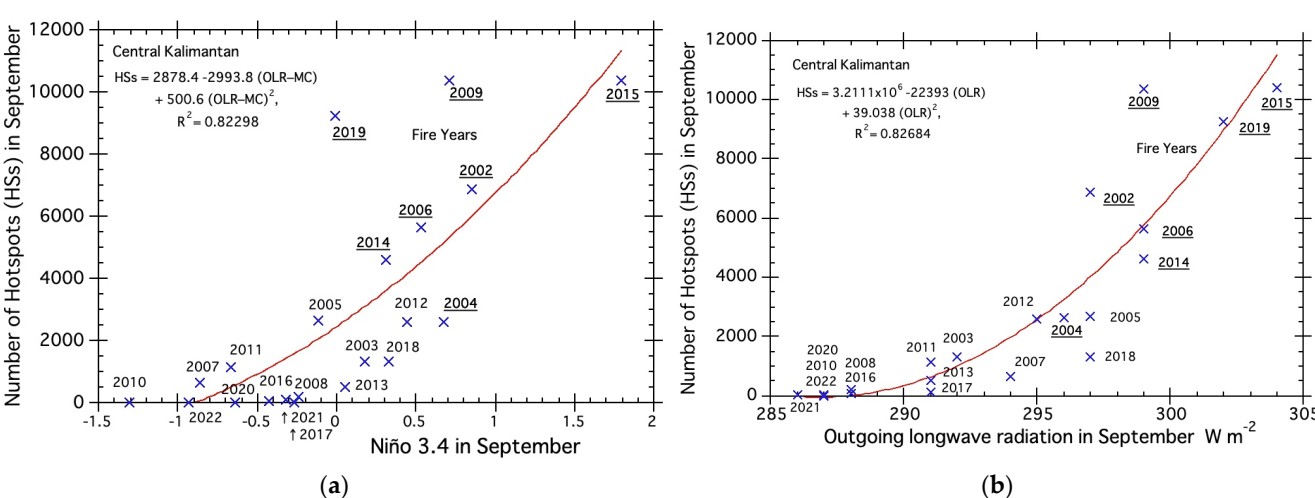

**Figure 8.** (**a**) Hotspots and Niño 3.4. (**b**) Hotspots and OLR, NOAA.

## 4. Discussion

This study was conducted to identify the fire weather conditions needed to assess future peat fires under climate change. Due to the lack of meteorological data, many studies have indirectly evaluated active peatland fire weather using meteorological indices, such as Niño 3.4, GWL, and SSTA. However, fire weather conditions need to be defined in order to assess future peat fires under climate change. In addition to this, the wide range of daily changes in tropical climates require hourly data for weather analysis. Fortunately, we were able to find and use hourly data from several locations in Indonesia to create this report. However, as there is only a list of hourly data in day-by-day data, the process of converting these data into consecutive daily data for analysis was time consuming. Nonetheless, analysis using limited 0.5-hourly data was able to clarify one important fire weather factor (land temperature) by comparing fire weather conditions during the active fire period (12 to 22 September) in 2015 (El Nino year) and 2021 (La Nina year).

### 4.1. Fire Distribution

Figure 1 was made using 21 years (2002–2022) of satellite-observed fire data (MODIS hotspot (HS)) to show very active fire areas (HSs > 100 yr.$^{-1}$) and weak fire areas. The highest number of HSs is 139 in northern Bahaur. The other three high HSs grid cells also located in the southern MRP Block-C (Figure 1). Four grid cells in southern central Kalimantan exceed 100 HSs yr.$^{-1}$. To clarify fire weather conditions there, we used satellite-observed long-term rainfall data at Bahaur (the nearest town name to describe potion). Weather 0.5-hourly data were obtained at the nearest weather station in Banjarbarue. Grid cells without numbers in Figure 1 show areas to protect from fires.

### 4.2. Fire Months and Rainfall

Figure 2a was made using 21 years (2002–2022) of satellite-observed fire data to show fire years and three fire months (August, September, and October). Figure 2b was made using 21 years (2002–2022) of rainfall data to show drought conditions of three fire months. Figure 2a,b show active fires in the top five years (2002, 2006, 2009, 2015, and 2019) occurred under severe drought conditions with monthly rainfall less than 50 mm.

### 4.3. Average Dry and Fire Season

By using relatively long-term (21 years (2002–2022)) rainfall data, we clearly defined average dry and wet seasons, such as shown in Figure 3a. We also find a very low rainfall

period (average rainfall of 1.4 mm) in the middle of dry season. This may be one of the reasons for very active fire areas in southern central Kalimantan.

Average fire season is also defined using relatively long-term (21 years (2002–2022)) fire (hotspot) data as shown in Figure 3b. Three fire periods defined from their activity (HSs day$^{-1}$) are surface fire, shallow peatland fire, and deep peatland fire [17]. The active fire period occurred during the deep peatland fire. These three fire activities were already explained by using ground water level (GWL) [17].

### 4.4. Various Fire Weather Conditions

We clarified important weather factors for central Kalimantan by comparing various fire weather conditions during active fire periods in 2015 (El Niño year) and the same period in 2021 (La Niña year). Analysis using diurnal weather data every 30 min from midnight (12:00 am) clarified various fire weather conditions during active fire days.

Various weather conditions during the active fire period in 2015 are summarized in Table 1. For comparison, weather conditions in 2021 and on 12 September 2015 are added in Table 1.

**Table 1.** Summary of weather conditions.

| Date, Month, Year | Temperature | | RH | | Wind Speed | | Wind Direction Change | Remarks |
|---|---|---|---|---|---|---|---|---|
| | Highest | ΔT | Lowest | ΔRH | Highest | ΔWS | | |
| 12–22 September 2015 | 36.0 | 2.7 | 37.0 | −12.6 | 21.0 | 1.9 | (S)→SE | Figures 5a and 6a |
| 12–22 September 2021 | 31.1 | 1.6 | 45.8 | −8.0 | 12.5 | 2.2 | SE→E | Figures 5b and 6b |
| Difference | 4.9 | 1.1 | −8.8 | −4.6 | 8.5 | −0.3 | | |
| 12 September 2015 | 37.0 | 2.7 | 30.0 | 14.2 | 28.0 | 3.4 | (SE)→E | Figure 7a,b |

Fire weather conditions in Table 1 suggest strong sea breeze made by mainly land temperature as there are no large differences in sea surface temperature on 22 September in 2015 and 2021 were 301° K and 302° K, respectively. The land temperature is the main factor for peatland fires in Kalimantan.

### 4.5. Various Fire Related Index

We introduce outgoing longwave radiation (OLR) to evaluate land temperature. There is a good correlation between fires and OLR as shown in Figure 8b. It can be said that the strong correlation is due to the fact that sea breezes became stronger when the land temperature became higher, and as a result, fires became more active. This index will be used to evaluate fire activities in Indonesia.

Finally, we hope that researchers will use the results of this report to help assess future peat fires and improve fire forecasting under climate change.

**Author Contributions:** Conceptualization, A.U. and H.H.; methodology, H.H.; software, A.U.; validation, A.U. and H.H.; formal analysis, A.U. and H.H.; writing—original draft preparation, A.U.; writing—review and editing, A.U. and H.H.; visualization. A.U. and H.H.; supervision, H.H. All authors have read and agreed to the published version of the manuscript.

**Funding:** This research received no external funding.

**Institutional Review Board Statement:** Not applicable.

**Acknowledgments:** We would like to thank (i) support for the Twentieth Century Reanalysis Project version 3 dataset is provided by the U.S. Department of Energy, Office of Science Biological and Environmental Research (BER), by the National Oceanic and Atmospheric Administration Climate Program Office, and by the NOAA Physical Sciences Laboratory; (ii) weather database from Weather Spark, Cedar Lake Ventures, Inc., 2500 Shadywood Rd Ste 510, Excelsior, MN 55331-6203, United States; and (iii) various maps from Google Map Pro by Google LLC. We also acknowledge the use of imagery from NASA's Worldview application (https://worldview.earthdata.nasa.gov (accessed on 11 March 2023)), part of NASA's Earth Observing System Data and Information System (EOSDIS).

**Conflicts of Interest:** The authors declare no conflict of interest.

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
