# Peer review of "Peatland Fire Weather Conditions in Central Kalimantan, Indonesia"

_fire, doi:10.3390/fire6050182_

Round 1

Reviewer 1 Report

Dear Authors,

Thanks for your works. Please find my enclosed comments as attached.

Best.

Author Response

Dear Reviewer 1

Thank you very much for your kind comments and suggestions:

General Comments:

(GC)

The manuscript provides an interesting study on the fire weather in tropical peatland, based on observed climate data in Kalimantan. Although the findings have a merit, but the manuscript in the current form consisted of some inconsistencies and coherency issues that makes it not suitable for publication. Substantial improvements on the manuscripts are still needed, especially in the introduction and method sections, and I expect to have a discussion on the limitation/weakness of the study. Further, proofread on the manuscript is necessary for its readability.

Answer: We revised as possible as we could.

  • Research gaps being addressed are rather weak. Just has not been clarified (L49) and listed previous studies may not substantially address the gaps (L49-64). Authors stated a similar analysis with another region in Indonesia. So, what is the novelty of this manuscript, then?

Answer: This report will be a first report to determine fire weather conditions during active fire period in Kalimantan. So far, no one has revealed this. Because it is very difficult to find any usable data. But we fortunately find it and could determine fire weather conditions.

Peat fire weather conditions are different place by place. For instance, active peatland fires in North Sumatra tended to occurred from January to March. But, active fires in Kalimantan occur from August to October. So, this different active fire season of both places are suggesting there are different fire weather conditions for winter and summer fires. So, results of both places have their own novelty.   

  • Authors need to restructure the manuscript in some places to improve the story telling of the manuscript. For instance, some sentences were misplaced such as L67-72 are better placed in method section not the introduction.

Answer: Revised

  • In the current form, replicability for the method is an issue, such as how to define the fire year, what is the assumption used for the analysis, how to deal with different time/spatial scale resolution of the data, and how to determine trend, etc.

Answer:

“Fire years”: “Fire year” is defined as the year in which about twice as many fires occurred as the average. We just use “Fire year” in Figure 2 (b) and Figure 8 for simple comparison with rainfall, Niño 3.4, and OLR. There is no reproducibility issue.

“Assumptions”: There is no special assumptions in this manuscript.

“Resolution of the data”: Main analysis of weather carried out using 30 minutes data. Other weather and fire data are mostly daily. But those differences have no effect on the important results.

“determine trend”: Fire and rainfall trend are mainly determined using accumulated curves.

  • Authors used different time scales for data rainfall, hotspots and weather data. how to deal with it. Also, different in data spatial resolution. Paragraph on this issue will improve the replicability of the paper.

Answer: As mentioned above, there is no special issues. Reproducibility issues also do not exist.

  • Some important definitions are missing, such as fire weather and fire year.

Answer: Fire year is defined in the manuscript.

“Fire weather”: Definition is “The types of weather that create favorable conditions for the start and spread of wildfires are collectively referred to as fire weather”.

  • Citation style is a big issue.

Answer: Corrected.

Specific comments:

Abstract:

  • L14, I wonder why authors used ‘preliminary analysis’ terminology here.

Answer: Deleted it. It comes from we have to use only limited weather data.

  • L21, I think researchers would not use other works in this case. More general recommendation for policy makers or future research will help the readers on the significant contribution of the research to the community.

Answer: Revised.

Method:

  • Fire distribution, does it represent fire regime in the study site?

Answer: Yes, fire distribution in figure 1 shows different fire regimes for MRP with high number of hotspots area and the Sebangau National Park with low number.

  • What is the justification the use of weather data from Banjarbaru as the Ex-MRP is far away from it (> 100Km)

Answer: There is no choice. Banjarbaru is only one site in central Kalimantan has weather data of every 30 minutes.

  • Definition on fire year is not found in any method, but it is in the results section.

Answer: we defined fire year.

  • L98, canal is not for irrigation (L98) but for drainage water.

Answer:  Yes, it is. However, the "canal" was built as an irrigation system to supply water to rice fields, but failed, as you may know.

  • Figure 1: please provide the period of year, and why the grid cells were in 1 degree resolution?

Answer:  It is 0.1 degree resolution. Corrected.

  • L119, Papua? We talked about Borneo

Answer:  It is Kalimantan. Corrected.

  • L131 what is ‘now’?

Answer:  It is present. Corrected.

  • Did any rational for the use of long term means for 1991-2020.

Answer:  It is often used to evaluate weather conditions. “long term means” could show anomaly.

  • Did any bias correction for weather data for Bahaur? Or it just uses as it is.

Answer:  No correction is made.

  • L148 used to show?

Period is deleted. Corrected.

  • Authors used rainfall pattern to determine drought condition, any reference on this? Why not use rainfall magnitude? I think from Figure 4, drought is defined based on the slope of cumulative rainfall. Authors may use polynomial equation to do this.

Answer:  As shown in Figure 4(a), the slope of cumulative rainfall is almost zero under a strong drought like in 2015. There is no need to use “rainfall magnitude”.

Results and discussion

  • Fire year?

Answer: Defined.

  • What are bold lines in Figures 3-4

Answer: They are cumulated curves of rainfall and hotspots.

  • Does trend analysis have been explained in Method?

Answer: “trend” added in 2.3. Analysis Methods

  • L397-430. Meaningless explanation for discussion section

Answer: Deleted. They are summarized in the table.

  • Discussion provides authors insight on the topic, how it differs from other studies, how the assumptions used influence the conclusions, and what new insight/novelty of the research.

Answer: We revised as possible as we could.

Citation

  • Citation type/style is not consistent throughout references. For example, variation in the position of year in the citation

Answer:  Corrected.

  • I found many grey literatures on the reference list, please provide the published peer review papers.

Answer:  All of our papers are peer-reviewed.

  • All of citation are listed in the introduction, which makes the richness of discussion is lack.

Answer: 

  • L39-41 needs citation (s).

Answer: citation is added.

  • The use of pronoun ‘We’ in L59, to whom it refers to (all authors of this manuscript)?

Answer: Yes. All authors of this manuscript.

Reviewer 2 Report

General Comments

The paper aims to identify the factors of “fire weather” in Central Kalimantan. The authors used various data sources (station and remote sensing) to determine that temperature, relative humidity, and wind played vital roles in promoting conditions that lead to peat burning. Some parts of the paper were unclear and required further clarification and analysis. Based on my observation, the paper needs to be revised before publication.

Methodology

Line 100: The disturbances mentioned, such as the canal, are the reasons for the fire in the study area. Can the authors cite references that support this?

Line 106: Please define, in this context, the definition of “hotspots.”

Line 111: The MRP+ contributes greatly to what?

Figure 1: The grid cells are not 1 by 1 deg but 0.1 by 0.1 deg.

Figure 1: There seems to be more than one block for “B:” one at the top of the map and the other at the bottom. Plus, a block for “Bb” was not explained. Please confirm the labels.

Figure 1: The weather station location should be labeled in the figure.

Line 119: How does the MODIS data used at Papua relate to this study?

Results

Line 180 and Figure 1: The yellow lines mentioned in this line are not visible in Figure 1.

Line 222: Why does the surface need to be dried, since the authors suggest it should be?

Figure 3: Specify the aggregate used to combine the data from 2002 to 2022.

Figure 4: In (a), the red line denotes 2015, but in (b), it means 2021. Please make the color scheme consistent.

Line 256: The authors repeatedly mention that the half-hourly data started from midnight in the methodology, results, and discussion. What is the significance of mentioning this?

Line 345: Typo for the word “places.”

Discussion

Section 4.4: Although it is more comprehensive to see the results in the list form, e.g., T1-T5, R1-R5, W1-W5, it will be more readable if they are summarized in a table. Furthermore, it is not apparent what the abbreviations T, R, and W mean.

Lines 440-444: More analyses and discussion are needed to convince the reader that OLR is more accurate in predicting fire than the conventional methodology.

Lines 434-435: The comparison to the study at Papua is abrupt. Please add more context so that it is more relevant to the study. This study was also mentioned in the other parts of the paper.

Spelling and Grammar

The objective states that weather data (T, RH, U, and WD) were used to “clear fire weather conditions.” Can the authors clarify the meaning of clearing the condition? I believe there is a misunderstanding of the word “clear.”

Multiple spelling errors and misuse of words were found in the manuscript, such as “bule,” “sever,” “minuet,” etc.

Author Response

Dear Reviewer 2

Thank you very much for your comments and suggestions:

General Comments

 The paper aims to identify the factors of “fire weather” in Central Kalimantan. The authors used various data sources (station and remote sensing) to determine that temperature, relative humidity, and wind played vital roles in promoting conditions that lead to peat burning. Some parts of the paper were unclear and required further clarification and analysis. Based on my observation, the paper needs to be revised before publication.

Answer: We revised manuscript.

Methodology Line 100: The disturbances mentioned, such as the canal, are the reasons for the fire in the study area. Can the authors cite references that support this?

Methodology Line 100:

Answer:  The term "canal" is used to distinguish between irrigation and drainage.

If the canal acts as an irrigation channel, it will provide sufficient water to suppress fires.

Conversely, when it acts as a drainage channel, the peat dries out and is more prone to fire.  

High hotspots values for MRP areas in Figure 1 and many studies in Reference [1,2,4..and so on] support the reasons for the fire.

Line 106: Please define, in this context, the definition of “hotspots.”

Answer:  It is number of hotspots.

Line 111: The MRP+ contributes greatly to what?

Answer:  This means: Most fires occurred in the area of ‘MRP+’.

We changed sentences.: This suggests that peatland fires existing in MRP+ contribute greatly as the total number HSs of top seven fires in MRP+ was 77,446, about 80% of the total number HSs (97,423) for 20 years from 2002.

Figure 1: The grid cells are not 1 by 1 deg but 0.1 by 0.1 deg.

Answer:  Corrected.

Figure 1: There seems to be more than one block for “B:” one at the top of the map and the other at the bottom. Plus, a block for “Bb” was not explained. Please confirm the labels.

Answer:  “Bb” stands for Banjarbaru, weather station like shown in the bottom in Figure 1.

We revised Figure 1.

Figure 1: The weather station location should be labeled in the figure.

Answer:  Corrected.

Line 119: How does the MODIS data used at Papua relate to this study?

Answer:  It is Kalimantan. Corrected.

Results Line 180 and Figure 1: The yellow lines mentioned in this line are not visible in Figure 1.

Answer:  Corrected: They are light yellow rectangles.

Line 222: Why does the surface need to be dried, since the authors suggest it should be?

Answer:  For peat to burn, the surface must be dry.

Figure 3: Specify the aggregate used to combine the data from 2002 to 2022.

Answer:  These data are just average daily data of rainfall and hotspots.

Figure 4: In (a), the red line denotes 2015, but in (b), it means 2021. Please make the color scheme consistent.

Answer:  Color corrected.

Line 256: The authors repeatedly mention that the half hourly data started from midnight in the methodology, results, and discussion. What is the significance of mentioning this?

Answer:  Time interval is different place by place.

Line 345: Typo for the word “places.”

Answer:  Corrected.

Discussion Section 4.4: Although it is more comprehensive to see the results in the list form, e.g., T1-T5, R1-R5, W1-W5, it will be more readable if they are summarized in a table. Furthermore, it is not apparent what the abbreviations T, R, and W mean.

Answer:  We make Table 1 and summarized them in Table 1.

Lines 440-444: More analyses and discussion are needed to convince the reader that OLR is more accurate in predicting fire than the conventional methodology.

Answer:  OLR is one of indicators to evaluate rainfall cloud activity and is used to evaluate rainfall. In this report, it is introduced as an indicator for evaluating ground surface temperature. Supplementary explanations have been added, and two references have been added.

Lines 434-435: The comparison to the study at Papua is abrupt. Please add more context so that it is more relevant to the study. This study was also mentioned in the other parts of the paper.

Answer:  To avoid confusion, deleted them.

Spelling and Grammar

The objective states that weather data (T, RH, U, and WD) were used to “clear fire weather conditions.” Can the authors clarify the meaning of clearing the condition? I believe there is a misunderstanding of the word “clear.”

Answer:  Revised.

Multiple spelling errors and misuse of words were found in the manuscript, such as “bule,” “sever,” “minuet,” etc. “minuet,” could not find

Answer:  Corrected.

Reviewer 3 Report

The manuscript “Peatland Fire Weather Conditions in Central Kalimantan, Indonesia” represents an assessment of meteorological conditions during fire events in the peatlands in Central Kalimantan (Indonesia). The comments for potential improvement of the manuscript are provided below.

Abstract

Abstract, the first sentence, “.... globally significant environmental impacts”? what are the facts for this conclusion?

“The objective of this study is to determine the fire weather needed to assess future peat fires under climate change.” You rather determined meteorological conditions during fire events, nothing more nothing less.

“Relatively large difference in temperature increase rates from 7:30 am in 2015 and 2021, were 2.7 and 1.6° C h-1 respectively”. Unclear writing.

Please, write the air temperature unit correctly (°C).  Generally, throughout the paper, specify which temperatures you are writing about (i.e., air temperature).

I suggest rewriting the abstract, the authors must clearly and precisely write the research topic, the used methodology and the most important results.

Keywords

The choice of keywords is inappropriate and chaotic. For some of these, there is no reason or logic why they are chosen.

Introduction

The authors put the sign of equality between meteorological conditions and fires. Are the observed fires solely the result of weather conditions? What exactly are the causes of fires?  Is there an influence of human activity? According to your claims, the cause of the fire is weather conditions. Or, the weather is favorable factor for the development and spread of fire?

Line 60-62, “-” is not correct sign.

Materials and Methods

It is so hard to follow your working algorithm for the used data set and to find any logic in them. Too many datasets, different spatial and temporal resolutions, without any explanation of  how related this data with adequate background in literature and studied topic.

Line 88-89. “The study area in Kalimantan (Borneo) covers the latitude range from 1.5° to 3.5° S (south latitude) and the longitude range from 113° to 115° E (area is about 49,500 km2) as shown in Figure 1”. Line 93. “Peatlands in Kalimantan cover about 57,600 km2 and are equivalent in area to that of Sumatra”.  The bold numbers (surfaces in km2) create confusion.

I suggest that you divide the content of Figure 1 into several figures to make presented content readable and easier to follow.

In “Analysis Methods” – methodology is missing (e.g. how exactly were defined dry and wet season, …., how exactly were defined “fire season, fire activities, etc., …..). There is a lot of confusion.

Results

The results are very difficult to follow. As stated above, if you do not explain well the data and methodology used it is very difficult to track what you got, moreover to see if the results were performed in the right way. Also, please find a better way to systematize the large amount of data you got as results (use more graphics, tables, …). Additionally, there is almost no citation in the results and discussion sections. Results are discussed without any connection to scientific frameworks.

A major revision is recommended.

Author Response

Dear Reviewer 3

Thank you very much for your kind comments and suggestions:

Abstract

Abstract, the first sentence, “.... globally significant environmental impacts”? what are the facts for this conclusion?

Answer: Peatland fires emit large amounts of greenhouse gases and have major environmental impacts.

“The objective of this study is to determine the fire weather needed to assess future peat fires under climate change.” You rather determined meteorological conditions during fire events, nothing more nothing less.

Answer: This report will be a first report to determine fire weather conditions during active fire period in Kalimantan. So far, no one has revealed this. Because it is very difficult to find any usable data. But we fortunately find it and could determine fire weather conditions.  

“Relatively large difference in temperature increase rates from 7:30 am in 2015 and 2021, were 2.7 and 1.6° C h-1 respectively”. Unclear writing.

Answer: Corrected as follows.

Relatively large difference in temperature increase rates from 7:30 am were 2.7 ℃ h-1 in 2015 and 1.6 ℃ h-1 in 2021.

Please, write the air temperature unit correctly (°C). Generally, throughout the paper, specify which temperatures you are writing about (i.e., air temperature).

Answer: Corrected

I suggest rewriting the abstract, the authors must clearly and precisely write the research topic, the used methodology and the most important results.

Answer: Revised

Keywords

The choice of keywords is inappropriate and chaotic. For some of these, there is no reason or logic why they are chosen.

Answer: They are all related to evaluate fire activities in Central Kalimantan.

Introduction

The authors put the sign of equality between meteorological conditions and fires. Are the observed fires solely the result of weather conditions? What exactly are the causes of fires? Is there an influence of human activity? According to your claims, the cause of the fire is weather conditions. Or, the weather is favorable factor for the development and spread of fire?

Answer: Large scale fires, regardless of the cause of the fire, are affected by the weather after they occur. Many large fires occur under high winds, and this study analyzes the weather conditions that produce high winds.

Line 60-62, “-” is not correct sign.

Answer: Revised。

Materials and Methods

It is so hard to follow your working algorithm for the used data set and to find any logic in them. Too many datasets, different spatial and temporal resolutions, without any explanation of how related this data with adequate background in literature and studied topic.

Answer: Revised。

Line 88-89. “The study area in Kalimantan (Borneo) covers the latitude range from 1.5° to 3.5° S (south latitude) and the longitude range from 113° to 115° E (area is about 49,500 km2) as shown in Figure 1”. Line 93. “Peatlands in Kalimantan cover about 57,600 km2 and are equivalent in area to that of Sumatra”. The bold numbers (surfaces in km2) create confusion.

Answer: You are misunderstood.

To reduce misunderstandings, we deleted "are equivalent in area to that of Sumatra " and made simple sentences.

Kalimantan has approximately 57,600 km2 of peatlands, with 30,100 km2 in Central Kalimantan alone. Figure 1 shows main part of peatland in Central Kalimantan.

I suggest that you divide the content of Figure 1 into several figures to make presented content readable and easier to follow.

Answer: Figure 1 was revised.

In “Analysis Methods” – methodology is missing (e.g. how exactly were defined dry and wet season, …., how exactly were defined “fire season, fire activities, etc., …..). There is a lot of confusion.

Answer: Revised them as possible.

Results

The results are very difficult to follow. As stated above, if you do not explain well the data and methodology used it is very difficult to track what you got, moreover to see if the results were performed in the right way. Also, please find a better way to systematize the large amount of data you got as results (use more graphics, tables, …). Additionally, there is almost no citation in the results and discussion sections. Results are discussed without any connection to scientific frameworks.

Answer: Revised contraction of manuscript.

A major revision is recommended.

Answer: Revised them as possible.

Round 2

Reviewer 2 Report

No further comments. 

Author Response

Dear Reviewer 2

Many thanks for your correction and sugestion, especially for English language and style is already checked.

Reviewer 3 Report

Comments:

Line 13- different data sets were used to study this issue.

Line 14-15 – Based on the meteorological data for the period of active fires (11 days), a large increase in air temperatures was observed due to sea breezes that began blowing in the morning.

Line 68 –the data from different data providers were used.

67-68 – “Fortunately, we found the data on the Internet. “. Please remove this sentence.

Line 68 – Weather data in meteorological data

Sentence in lines 59-63 remove in the next paragraph.

Line 82 – 10 m above sea level

Line 116– check numbers for geographical coverage.

Line 125 – What is “the weather underground”?

Line 128. Niño 3.4 data

Line 108. “pixel resolution of 1 km.” and line 143 “the 0.1 degrees grid cell (0.1° x 0.1°, latitude and longitude)”, check these numbers.

Line 242 – from the 12th to the 22nd of September

Line 260. Use correct sign for “minus”, check through the entire paper.

Author Response

Dear Reviewer 3:

Thank you for your comments and final checks.

Comments:

* Line 13- different data sets were used to study this issue.

(A1): "diurnal" is inserted in line 13.

“the only available diurnal weather data was used to analyze fire weather”.

We used only one diurnal weather data.

* Line 14-15 – Based on the meteorological data for the period of active fires (11 days), a large increase in air temperatures was observed due to sea breezes that began blowing in the morning.

(A2): Revised. Thank you for your correction.

* Line 68 –the data from different data providers were used.

(A3):  "diurnal" is inserted in line 65

“Only one site in central Kalimantan has diurnal weather data (air temperature, relative humidity, wind speed, and wind direction) of every 30 minutes from 12 am to 11:30 pm.”

* 67-68 – “Fortunately, we found the data on the Internet. “. Please remove this sentence.

(A4): Deleted.

* Line 68 – Weather data in meteorological data

(A5): We used only one diurnal weather data.

* Sentence in lines 59-63 remove in the next paragraph.

(A6): "lines 59-63" moved to “2.3. Analysis Methods”.

* Line 82 – 10 m above sea level

(A7): Corrected.

* Line 116– check numbers for geographical coverage.

(A8): All numbers confirmed.

* Line 125 – What is “the weather underground”?

(A9): It is the web site name and suggest weather data measured at ground level.

"underground” may come from "wunderground" (https://www.wunderground...).

* Line 128. Niño 3.4 data

(A10): Corrected.

* Line 108. “pixel resolution of 1 km.” and line 143 “the 0.1 degrees grid cell (0.1° x 0.1°, latitude and longitude)”, check these numbers.

(A11): They were checked. The pixel resolution of MODIS hotspot data is defined as 1 km.

* Line 242 – from the 12th to the 22nd of September

(A12): Revised.

* Line 260. Use correct sign for “minus”, check through the entire paper.

(A13): Checked.
